# A Study of Brain Function Characteristics of Service Members at High Risk for Accidents in the Military

**DOI:** 10.3390/brainsci13081157

**Published:** 2023-08-02

**Authors:** Sung-Oh Choi, Jong-Geun Choi, Jong-Yong Yun

**Affiliations:** Department of Protection and Safety Engineering, Seoul National University of Science and Technology, Seoul 01811, Republic of Korea

**Keywords:** stress, psychological load, accident prevention, brain waves, pulse waves, military death, military misfits, identify incident risks

## Abstract

Military accidents are often associated with stress and depressive psychological conditions among soldiers, and they often fail to adapt to military life. Therefore, this study analyzes whether there are differences in EEG and pulse wave indices between general soldiers and three groups of soldiers who have not adapted to military life and are at risk of accidents. Data collection was carried out using a questionnaire and a device that can measure EEG and pulse waves, and data analysis was performed using SPSS. The results showed that the concentration level and brain activity indices were higher in the general soldiers and the soldiers in the first stage of accident risk. The body stress index was higher for each stage of accident risk, and the physical vitality index was higher for general soldiers. Therefore, it can be seen that soldiers who have not adapted to military life and are at risk of accidents have somewhat lower concentration and brain activity than general soldiers, and have symptoms of stress and lethargy. The results of this study will contribute to reducing human accidents through EEG and pulse wave measurements not only in the military but also in occupations with a high risk of accidents such as construction.

## 1. Introduction

### 1.1. Background and Purpose of the Study

Casualties in the military are not only demoralizing but also undermine public confidence in the military. However, the South Korean military is currently a conscription system, meaning that all young men with certain qualifications are enlisted regardless of their will. Therefore, the military has implemented various efforts to prevent casualties, such as operating counselors, developing suicide prevention education programs, and establishing safety systems for each discharge, but they have not led to a decrease in casualties [1,2,3].

According to a 2022 military release [4], there are more than 80 casualties per year, increasing public anxiety and reducing the military’s combat capabilities, and there is an urgent need to prevent military casualties.

The military has analyzed casualty incidents and assessed that although there are various causes, severe stress, anxiety, lethargy, and depression among soldiers contribute to human errors and suicides [5,6,7,8,9,10,11]. Among individual characteristics, research on stress has shown that stress is an important catalyst for psychiatric disorders such as depression and anxiety [12,13,14,15,16]. Stress-induced negative emotions lead to unclear decision-making, poor attention, and accidents [17], and there has been a 60–80% correlation between stress and accident rates [18,19,20,21,22].

While there have been many studies using EEG and pulse waves to measure stress and depression in the general population [23,24,25,26], there is a lack of research on the relationship between EEG and pulse waves and the psychological state of military personnel [27,28,29,30,31].

Therefore, the purpose of this study is to analyze and present the differences in brain function and autonomic nervous system between general soldiers who are well-adjusted to military life and soldiers who are not well-adjusted to military life and are at risk of accidents. Therefore, EEG and pulse waves were measured and analyzed to determine the differences in terms of four EEG indices and five pulse wave indices.

### 1.2. Brain Function and Autonomic Nervous System Characteristics

Brain waves are electrical signals composed of multiple oscillatory components that result from the complex interaction of excitatory and inhibitory neurons and can be categorized by frequency band [32,33]. Brain waves are categorized into five types in order of decreasing frequency: delta (δ) waves from 1 to 4 Hz, theta (θ) waves from 4 to 8 Hz, alpha (α) waves from 8 to 13 Hz, beta (β) waves from 13 to 30 Hz, and gamma (γ) waves from 30 to 50 Hz [34,35,36]. 

In a resting EEG with eyes closed, natural oscillations reflecting an idle cortical state become dominant, with the dominant peak frequency typically located in the 4–13 Hz band. Previous reports have observed that the dominant oscillation frequency, which typically appears in the alpha band during normal aging, is lower in patients with cognitive impairment [37,38,39,40].

Pulse waves are the waves formed by the pulse as it travels to the peripheral nerves, and are the wavelengths at which blood propagates in waves from the heart [41,42]. Pulse wave measurements can be used to test the function of the autonomic nerves that directly control the heartbeat. The autonomic nervous system is composed of sympathetic and parasympathetic nerves, which act antagonistically to each other [43,44,45]. In stressful situations, the sympathetic nerves, which are responsible for attack/defense mechanisms, are relatively more active than the parasympathetic nerves, which are responsible for restoration/relaxation mechanisms [46]. In 1981, power spectral analysis was introduced as a way to quantitatively assess parasympathetic and sympathetic control of the heart and to quantify the balance of the autonomic nervous system, allowing for more detailed functional testing [41].

The five pulse wave indices that can be measured through pulse waves are mainly used to measure and analyze physical stress and autonomic nerve health through PPG (Photoplethysmography).

### 1.3. Stress, Depression, and Their Relationship to Brain Function and the Autonomic Nervous System

Preclinical and clinical studies have demonstrated that stress or depression can lead to atrophy and cell loss in limbic brain structures that are critically involved in depression [47,48,49,50,51,52,53]. Convergent evidence over the past 20 years indicates that prolonged stress leads to overall neuronal atrophy and synaptic depression in the PFC and hippocampus [54,55,56,57], while regions such as the amygdala and NAc exhibit changes consistent with neuronal hypertrophy and synaptic potentiation [54,58,59]. The hippocampus provides inputs to other brain regions, including the prefrontal cortex, cingulate cortex, and amygdala, which contributes heavily to altered mood and emotion in depression [47,60,61,62,63]. Many studies report reduced hippocampal and prefrontal cortex volumes in depressed patients [47,64,65,66,67,68,69].

Previous studies have shown that 20–40% of patients with depression have abnormal EEG findings [70,71,72], with asymmetries in EEG activity in frontal regions [73,74,75,76]. The prefrontal cortex is affected by depression, anxiety, and stress, and alpha waves in the prefrontal cortex tend to be less active in depressed patients than in normal people [77,78,79]. Researchers have also found an average decrease in alpha and theta waves and an increase in beta waves in the EEG of depressed patients [80,81]. Neurotransmitters such as serotonin and dopamine play a role in regulating mood and cognitive function. Stress also causes changes in beta waves, which are associated with attention and persistence among EEG indices [12,82,83,84,85]. Reduced noradrenaline (NE) release to target areas such as the prefrontal cortex and amygdala limits anxious behavior [86,87].

There is a causative relationship between human mood and physical diseases such as HT. Patients with CVD commonly have anxiety and depression [88,89]. A previous study in a heterogeneous population of all ages showed that anxiety, depression, and HT are clearly related [88,90,91]. Inflammatory cytokines are increased in depression, which may explain the serotonergic, noradrenergic, and dopaminergic dysfunction of depression [92,93,94,95].

Humans experience pulse wave changes when subjected to external stressful stimuli [96], mental stress has been proven to lead to changes in the autonomic nervous system and further affect cardiac activity [97,98,99,100,101], and many studies on stress and depression have used heart rate variability (HRV) analysis to detect mental state [102,103,104,105,106]. Furthermore, Kang et al. [107] have categorized several combinations of EEG waves into formulas that best represent stress states, using the ratio of theta and M-beta waves.

## 2. Previous Research

### 2.1. Characteristics of Military Organizations

Kwon In-hyuk (2004) [108] found that the military is internally conflicted due to the uniformity and closure of the military culture and its conflict with the democratic and open culture of the new generation of soldiers. It was found that the embedded conflict structure relaxes the military organization and disrupts the military discipline, leading to a weakening of combat power and causing accidents. In addition, undemocratic practices directly increase soldier stress and contribute to accidents.

### 2.2. Relationship between Stress, Depression, and Traumatic Events

The compensatory imbalance of job stress is the greatest risk factor for problems with psychiatric disorders, and while stress is not a direct cause of errors, it does lead to inappropriate strategies for dealing with stressful situations [109,110,111,112]. Stress, anxiety, and depression are important markers that can affect the brain, behavior, and cognition, and stress and depression have been linked to the occurrence of accidents [17,113,114]. Worker stress levels have been shown to have a negative impact on safety behavior [115,116,117], and it has been argued that reductions in occupational stress can improve safety compliance. Alonso et al. [117] reported a high prevalence of job stress and emotional exhaustion symptoms. Kaushik, P. [75] stated in a study that depression causes sadness and anhedonia, which leads to decreased work performance and productivity, and in extreme cases can even lead to suicide, which is the fourth leading cause of death among 15–29-year-olds.

## 3. Research Methods

### 3.1. Choosing Who to Study

The South Korean military categorizes health status through enlistment physicals and personality tests, and sends personnel in generally good health to the Forward Corps. Personnel who are at risk of accidents due to their inability to adapt to military life after enlistment are managed by risk level. Stage 1 risk soldiers with mild maladjustment are managed separately by their units, while stage 2 risk soldiers with severe maladjustment are convened by forward corps to receive two weeks of service adjustment training by experts, including character education and counseling. Tier 3 at-risk Soldiers who have determined that further military life is limited and are in the process of being discharged are convened by their forward corps and managed until discharge.

The study population was divided into two groups: one group of normal soldiers who were well-adjusted to military life and three groups of soldiers who were at risk of accidents. The personnel were selected from eight units of the Forward Corps (Gyeonggi-do and Gangwon-do, South Korea), including 136 general soldiers, 42 soldiers with a risk of an accident, 47 soldiers with a very high risk of an accident, and 26 soldiers with the highest risk of an accident. To minimize the effect of enlistment, we excluded individuals who were taking medication for illness before enlistment. Regular soldiers and Phase 1 soldiers were selected from the infantry, artillery, engineer, and information technology units under the Forward Corps. For the second and third-tier soldiers, we excluded two soldiers who were not willing to participate in the study from the accident risk group, which is convened and managed by the Forward Corps.

### 3.2. Scope and Procedure of the Study

The scope of this study was to determine the differences in brain function and autonomic nervous system between soldiers who are well-adjusted to military life and those who are not and are at risk of accidents through EEG and pulse wave measurements and questionnaires. The procedure of the study was as follows.

First, we measured the EEG and pulse waves of one group of normal soldiers and three groups of soldiers at risk for accidents.

Second, a survey was conducted among those whose EEG and pulse waves were measured.

Third, we conducted a statistical analysis using the collected data.

### 3.3. Measurement Equipment

For EEG and pulse wave measurements, we used the EEG 2-channel dry and non-invasive measurement device shown in Figure 1, developed by Laxa and OmniCNC. This equipment is a medical device that connects via Bluetooth communication and transmits measurement data to a tablet. This product was tested on 300 healthy adult volunteers aged 19 to 69 years old from December 2007 to March 2008 at the Clinical Research Center of Asan Medical Center in Seoul.

### 3.4. Data Collection and Analysis

#### 3.4.1. Brain Waves

Electroencephalography is a method of recording the electrical activity of cortical pyramidal neurons in the brain; therefore, when a large number of neurons are activated simultaneously, enough postsynaptic potentials can be generated to be detected from the scalp of the brain [118,119,120]. By comparing and analyzing EEG waves at different wavelengths, it is possible to determine the level of concentration level, psychological stress, imbalance of left and right brain activity, and brain activity. Scalp EEG methods are non-invasive, relatively safe and fast, inexpensive and widely available, and allow for numerous repeat measurements [120,121,122,123].

This study analyzed MDF, PF, and ATR from EEG data measured while sitting comfortably and with eyes closed [124].

The MDF measured the median frequency and was calculated in two steps.

All spectral power values in the 4–13 Hz frequency domain were summed and divided by 2.The frequency where the cumulative power in the 4–13 Hz frequency domain first exceeded the value calculated in step 1 was selected.

PF was determined as the frequency at which the EEG spectrum has the most power in the 4–13 Hz frequency range.

ATR measures the ratio of power in the alpha rhythm (8–12.99 Hz) to the theta rhythm (4–7.99 Hz).

#### 3.4.2. Pulse Wave

Heart rate variability (HRV) has been used in clinical practice to assess overall autonomic function and can be used to assess autonomic cardiovascular function and health, as well as monitor stress and fatigue [125,126,127]. HRV can be investigated non-invasively using photoplethysmography (PPG), which can detect changes in blood volume in microvascular beds, and the HRV index is known to be an important indicator of cardiorespiratory performance. The greater the HRV index, the lower and wider the distribution pattern, the healthier the person [128,129,130]. PPG pulses are used to measure pulse variability and assess the sympathetic and autonomic nervous systems, and waveform analysis can determine the values of various indicators related to the cardiovascular system. However, due to the difficulty of recognizing inflection points in the pulse, the second derivative of the PPG pulse (SDPPG) and TDPPG are used for detailed and objective evaluation [131,132,133].

In this study, the PPG device was used to measure heart rate pulses while sitting comfortably and with eyes closed. The PPG sensor system used visible light with a wavelength of 640 nm and a relative sensitivity of 60%. The sensor circuit unit converts current to voltage, amplifies and filters the signal, and provides the final output signal. The frequency response of the receiver (photodiode) is 0.3–5 Hz in the −3 dB band of the characteristic curve (gain with frequency). This frequency response of the filter removes unwanted noise. The PPG data were measured at a 250 Hz sampling frequency. To remove trends and drifts, the initial preprocessing step prioritized trend removal. Low- and high-frequency noise, such as traveling noise and breathing noise, was removed from the DC component using a fourth-order bandpass zero-phase Butterworth filter with a cutoff frequency of 0.4 to 9 Hz, accounting for phase delay and distortion [134,135,136]. For the morphologically volumetric pulse trajectory, the point where the notch is minimized after passing through the mid-vein notch was set as the end of the pulse. Immediately after is the start of the pulse and is defined by the pulse-to-pulse interval [137,138,139]. Representative single-pulse values were obtained from the values of the pulse-to-pulse interval by calculating the average of each time point relative to the pulse onset. The second and third derivatives were used to obtain SDPPG and TDPPG from the single-pulse PPG signal of each subject [133,140,141].

## 4. Group Design and Participants

Looking at the demographic characteristics of the study subjects, 94 (39.2%) were 20 years old, with one each of 18 and 32 years old. In terms of education, 171 (71.2%) were enrolled in college, and 58 (24.2%) had a high school diploma or less, including one middle school graduate. Other general characteristics of the study subjects, such as rank and physical fitness, are shown in Figure 2.

## 5. Data Collection and Analysis Approaches

All data collected in this study were processed using the statistical program SPSS 27.0, and the significance level of the statistics was set at 95% (*p* < 0.05). A one-way analysis of variance (ANOVA) was performed to calculate the mean, and standard deviation, and test for significant differences between groups. Then, a post hoc test was used to test whether there was a difference between the groups.

We also correlated the EEG and pulse wave measurements of 251 participants with the survey results of 240 participants, excluding 11 who refused to complete the survey. The analysis utilized Pearson correlation analysis (significance level < 0.05), a method used to determine the degree of linear relationship between variables measured on a quantitative scale (equal/ratio scale).

## 6. EEG, Pulse Waves, and Survey Results

### 6.1. Analyze the EEG Index of Four Populations

The EEG analysis analyzed “Concentration level”, which evaluates the interest in the surroundings and the ability to immerse oneself in a particular object; “Brain activity”, which evaluates the activation of brain cognitive activity at the level of mental workload; “Psychological stress”, which evaluates the level of brain tension and anxiety; and “Imbalance of left and right brain activity”, which checks the neurological balance of left and right brain waves.

First, we compared the means between the groups for each EEG index to check for differences, and used a one-way analysis of variance (ANOVA) to determine the probability of significance between the groups to determine if there was a significant difference. Then, post hoc tests (multiple comparisons) were performed on the EEG indices that showed significant differences between groups.

#### 6.1.1. Compare between-Group Means by EEG Index

As shown in Table 1, The “Concentration level” index, which assesses the ability to concentrate, was highest for general soldiers and lower for each accident risk level. This suggests that acclimatization to military life and concentration are correlated, and that poor acclimatization can lead to weaker concentration due to anxiety symptoms. “Brain activity”, which assesses the activation of brain cognitive activity, was higher for soldiers in accident risk 1 and the general population than for soldiers in accident risk 2 and 3. This suggests that soldiers who have a normal routine in their units have more active brain activity than soldiers who spend their days sitting still and attending training sessions in the forward corps. “Psychological stress”, which assesses the level of tension and anxiety in the brain, was highest for regular soldiers, followed by accident risk levels 2, 1, and 3. In general, you would expect to see a higher percentage of people who have not adjusted to military life, but due to the nature of the military organization, there is a higher percentage of soldiers who are responsible and follow a normal routine. “Imbalance of left and right brain activity”, a measure of the balance between left and right brain waves, was highest for soldiers in accident risk 1, followed by general soldiers, accident risk 2, and accident risk 3.

#### 6.1.2. Compare Groups by EEG Index

We tested the homogeneity of variance for each EEG index, and as shown in Table 2, the significance level is greater than 0.05 for three indices except for concentration, so the groups have the same variance. We then tested whether there were differences in the four EEG indices between the groups using an analysis of variance.

As shown in Table 3, The Concentration level index showed that the probability of significance (*p* = 0.001) of the test statistic (F = 6.093) was less than the significance level (α = 0.05), and the brain activity index showed that the probability of significance (*p* = 0.001) of the test statistic (F = 5.495) was less than the significance level (α = 0.05). However, the Psychological stress index showed that the probability of significance (*p* = 0.168) of the test statistic (F = 1.698) was greater than the significance level (α = 0.05), and the Imbalance of left and right brain activity index showed that the probability of significance (*p* = 0.469) of the test statistic (F = 0.848) was greater than the significance level (α = 0.05).

Therefore, the EEG showed that the Concentration level index was different between groups with unequal variance, while the brain activity index was different between groups with the same variance.

#### 6.1.3. Brain Wave Post-Test

Post hoc tests (multiple comparisons) were performed on the Concentration level and brain activity indices that differed between the four groups, including the concentration index, which had unequal variances in the homogeneity of variance test, to determine which groups differed.

As shown in Table 4, Concentration level indices with unequal variances were subjected to multiple comparisons using Dunnett T3. The results showed that the significance probabilities of normal soldiers, accident risk stage 2 soldiers, and accident risk stage 3 soldiers were 0.014 and 0.000, respectively, which were less than the significance level (α = 0.05), and the significance probability of accident risk stage 1 soldiers and accident risk stage 3 soldiers was 0.003, which showed a significant difference.

The Brain activity indices with the same variance were subjected to Scheffe’s multiple comparison. As a result, there was no significant difference between normal soldiers and accident risk 1 soldiers with a significance level of 0.981, but there was a significant difference between normal soldiers, accident risk 2 soldiers, and accident risk 3 soldiers with a significance level of 0.039 and 0.040, respectively.

Thus, of the four EEG indices, we can see that the Concentration level indices and Brain activity indices are significantly different between the groups.

### 6.2. Analyze Pulse Wave Indices for Four Populations

Pulse wave measurements were used to analyze heart health, body stress and cumulative fatigue, physical vitality, and autonomic nerve system health. “Heart health” is determined by analyzing heart rate variability (HRV). “Body stress” is the immediate body response to current stress. “Cumulative fatigue” analyzes weakened autonomic nerve function and decreased heart rate variability to determine the body’s fatigue due to repetitive stress. “Physical vitality” analyzes the ratio of sympathetic and parasympathetic nerves to determine your current level of vitality or lethargy. “Autonomic nervous system health” is the health of the autonomic nervous system, which is responsible for metabolism.

The test procedure was to compare the means of the groups by pulse wave index to check for differences, and to determine whether there was a significant difference between the groups using a one-way analysis of variance (ANOVA). Post hoc tests (multiple comparisons) were performed on the pulse wave indices that showed significant differences between groups.

#### 6.2.1. Compare Group Means by Pulse Wave Index

As shown in Table 5, The “Heart health” index, as measured by heart rate variability (HRV) analysis, was highest for regular soldiers. The results also showed similarities between normal soldiers living a normal life in the unit and those in the first stage of accident risk, and differences between those in the third and second stages of accident risk, which are managed separately by the corps. The “Body stress” index, which measures the immediate physical response to stress, showed the following results: accident risk 3, 2, 1, and normal soldier. This suggests that the psychological state of being under stress due to not adapting to military life is a physical reaction, as previous studies have shown. The “Cumulative fatigue” index, which checks the body’s fatigue due to repetitive stress, was higher for general soldiers living a normal life in the unit and for soldiers in the first stage of accident risk than for soldiers in the second and third stages of accident risk, who are separately convened and receive character education and counseling. The “Cumulative fatigue” index analyzes the weakening of autonomic nerve function and decreased heart rate variability, a phenomenon that occurs because normal military life is accompanied by constant physical fatigue and mental stress such as responsibility. The “Physical vitality” index, which determines how energized or helpless you feel at any given moment, was as follows: general soldier, accident risk 2, 1, and 3. This reflects the feelings of helplessness and depression that often accompany soldiers who are unable to adjust to military life. The “Autonomic nervous system health” index, which measures the health of the autonomic nervous system, showed the following results: general soldiers, accident risk 1, 3, and 2. We know that stress, depression, and anxiety affect the autonomic nervous system in soldiers at high risk for accidents.

#### 6.2.2. Compare Groups by Pulse Wave Index

We tested the homogeneity of variance of each pulse wave index, and as shown in Table 6, except for “Cumulative fatigue” and “Physical vitality”, the significance probability is greater than 0.05 for three indices, so the groups of three indices have the same variance. Then, the analysis of variance was used to test whether there is a difference in the five indices of pulse wave between the groups.

As shown in Table 7, The *p*-value for the “Heart health” index (F = 1.873), the *p*-value for the “Cumulative fatigue” index (F = 1.467), and the *p*-value for the “Autonomic nervous system health” index (F = 2.594) are all greater than the significance level (α = 0.05), indicating that there is no difference between the groups.

However, there was a statistically significant difference between the groups, as the test statistic for the “Body stress” index (F = 4.977) (*p* = 0.002) and the test statistic for the “Physical vitality” index (F = 3.289) (*p* = 0.021) were smaller than the significance level (α = 0.05) (*p* < 0.05).

Therefore, the “Physical vitality” index in the pulse wave was analyzed as having a difference between groups with unequal variance, while the “Body stress” index was analyzed as having a difference between groups with equal variance.

#### 6.2.3. Pulse Wave Post-Test

Post hoc tests (multiple comparisons) were performed on the “Body stress” and “Physical vitality” indices that showed differences between the four groups, including the “Physical vitality” index, which did not have equal variances in the homogeneity of variance test, to determine which groups differed.

As shown in Table 8, The “Body stress” index with the same variance was subjected to a Scheffe multiple comparison. As a result, there was no significant difference between general soldiers and soldiers in the first stage of accident risk with a significance rate of 0.953, but there was a significant difference between normal soldiers, accident risk tier 2 soldiers, and accident risk tier 3 soldiers with a significance level of 0.027 and 0.042, respectively. The “Physical vitality” indices with unequal variances were subjected to multiple comparisons using Dunnett’s *t* test. The result shows that there is a significant difference between general soldiers and accident risk stage 3 soldiers with a significance probability of 0.007.

Therefore, among the five indices of pulse wave, we can see that the “Body stress” and “Physical vitality” indices are significantly different between the groups.

### 6.3. Correlate EEG, Pulse Waves, and Survey Results

Of the 251 people whose EEG and pulse waves were measured, 240 were surveyed, excluding 11 who refused to participate. The questionnaire was fully explained beforehand so that the participants could fill out the questionnaire as honestly as possible about their psychological state. The questionnaire utilized a 24-item scale for categorizing types of maladjustment used by Liu (2014) [142]. Each item was scored on a 5-point scale ranging from “very much so” to “not at all”, with higher scores indicating closer to the corresponding scale.

Correlations were analyzed between EEG, pulse wave, maladjustment, and suicidal ideation. The analysis utilized Pearson’s correlation analysis (significance level < 0.05), a method used to determine the degree of linear relationship between variables measured on a quantitative scale (equivalence/ratio scale).

The means of the EEG, pulse wave, maladjustment, and suicidal ideation indices are shown in Table 9, with standard deviations ranging from 0.4 to 7.8.

#### 6.3.1. Analyzing the Correlation between EEG and Pulse Wave Indices

As shown in Table 10, The brain activity index, psychological stress index, and imbalance of left and right brain activity index in the EEG were not correlated with any of the five indices in the pulse wave at a significance level of 0.085 or higher. However, the concentration level index was correlated with the autonomic health index in the pulse wave at a significance level of 0.034.

#### 6.3.2. Correlating Brainwave Indices with Survey Results

As shown in Table 11, Among the EEG indices, the concentration level index correlates with the maladjustment scale and the suicidal ideation scale at 0.002 and 0.001, respectively. The brain activity index is also correlated with the maladjustment scale at 0.002 and with the suicidal ideation scale at 0.000.

#### 6.3.3. Correlate Pulse Wave Indices with Survey Results

As shown in Table 12, Among the indices measured by pulse wave, the physical stress index correlates with the maladjustment scale and the suicidal ideation scale at 0.002 and 0.000, respectively. In addition, the physical vitality index correlated with the maladjustment scale at 0.005 and the suicidal ideation scale at 0.043.

## 7. Conclusions

Neurotransmitters and specific synaptic receptor activation influence stress-related behavioral and physiological responses [143,144,145,146,147]. In addition, many studies have shown that psychological conditions such as stress, depression, and anxiety are linked to physical health, including cardiovascular disease [89,148,149,150,151]. Advances in neuroimaging and NIBS techniques have enabled researchers to explore the underlying mechanisms of disorders such as depression, SCZ, anxiety, and post-traumatic stress disorders [152,153,154,155]. Identifying the neural circuits and networks involved in these disorders can lead to targeted interventions that aim to modulate brain activity and restore normal function [152,156]. Therefore, if it is possible to check the psychological state using brain waves and pulse waves, it will contribute to preventing suicide accidents and safety accidents caused by human error.

The results of the study showed differences in brain function and autonomic nervous system between normal soldiers and soldiers maladjusted to military life. and that EEG and pulse waves are correlated and reliable when compared to survey results. These results do not take into account all the different conditions, such as the health of the individual and the environment. Furthermore, there are many contradictions when it comes to psychological health based on just a few indices of EEG and pulse waves. However, it is significant that the military maladjusted soldiers generally showed lower levels of concentration and brain activity in the EEG and higher levels of physical stress, and lower levels of physical vitality in the pulse wave.

The study focused solely on determining whether there were differences in EEG and pulse waves between soldiers who were well-adjusted to military life and those who were not and were at risk of accidents. In the future, further research is needed on the correlation between EEG and pulse wave indices and the correlation of EEG and pulse wave measurements with the military’s current character and relationship type tests. Further research is needed on how to manage identified accident-risk soldiers by applying neurofeedback training.

There have been countless studies of EEG and pulse waves in civilian populations. However, there have been very few studies on military personnel. This is due to the nature of the military, which makes it difficult to get permission to study soldiers, and even when permission is granted, it is difficult to study soldiers who are spread out across the country. The significance of this study is that it was conducted among soldiers across the country, and in order to be more precise, it was divided into three groups of soldiers who were unable to adjust to military life based on their psychological state. In addition, the data analysis is presented in an average-oriented manner to make it easier for anyone to understand, even those who do not have expertise in EEG and pulse waves.

We believe that in the future, more military personnel and academics will be interested in neuroscience and contribute to its research.

## Figures and Tables

**Figure 1 brainsci-13-01157-f001:**
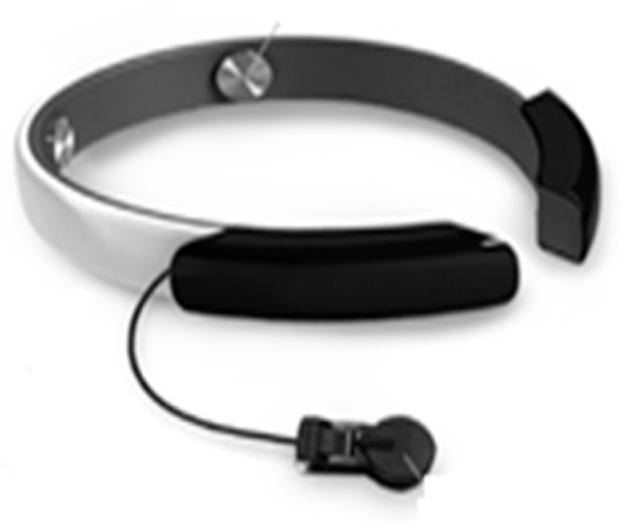
Omnifit headset.

**Figure 2 brainsci-13-01157-f002:**
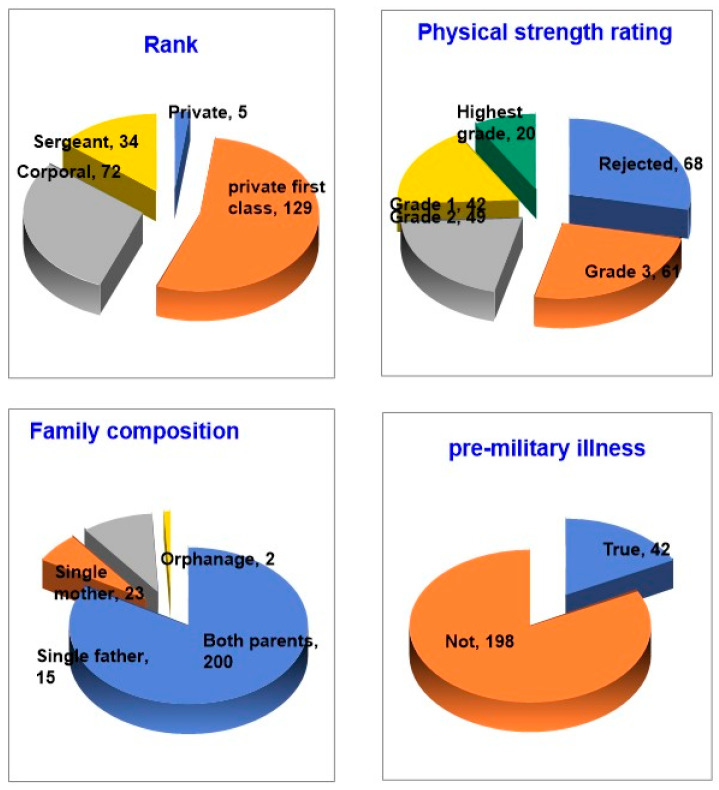
General characteristics of research subjects.

**Table 1 brainsci-13-01157-t001:** Technical Statistics.

		N	Average	Standard Deviation	Standard Error	95% for the Mean Confidence Interval		Minimum	Maximum
						Lower Limit	Upper Limit		
Concentration Level	General soldiers	136	6.555	2.0031	0.1718	6.215	6.895	2.0	10.0
	Accident risk level 1 soldier	42	6.517	2.0204	0.3118	5.887	7.146	3.2	9.9
	Accident risk level 2 soldier	47	5.649	1.5998	0.2334	5.179	6.119	2.6	9.8
	Accident risk level 3 soldier	26	5.169	0.9768	0.1916	4.775	5.564	3.5	7.2
	Subtotal	251	6.235	1.9116	0.1207	5.998	6.473	2.0	10.0
Brain activity	General soldiers	136	77.43	7.883	0.676	76.10	78.77	43	93
	Accident risk level 1 soldier	42	78.00	8.175	1.261	75.45	80.55	61	95
	Accident risk level 2 soldier	47	73.68	6.763	0.987	71.70	75.67	52	87
	Accident risk level 3 soldier	26	72.69	6.638	1.302	70.01	75.37	57	81
	Subtotal	251	76.33	7.821	0.494	75.36	77.31	43	95
Psychological stress	General soldiers	136	6.096	1.2835	0.1101	5.878	6.313	3.7	10.0
	Accident risk level 1 soldier	42	5.707	1.0086	0.1556	5.393	6.021	4.2	8.2
	Accident risk level 2 soldier	47	5.870	1.2360	0.1803	5.507	6.233	3.8	8.9
	Accident risk level 3 soldier	26	5.688	1.0203	0.2001	5.276	6.101	4.0	8.5
	Subtotal	251	5.946	1.2130	0.0766	5.795	6.097	3.7	10.0
Imbalance of left and right brain activity	General soldiers	136	5.007	0.6024	0.0517	4.905	5.110	3.0	7.0
	Accident risk level 1 soldier	42	5.167	0.4897	0.0756	5.014	5.319	5.0	7.0
	Accident risk level 2 soldier	47	5.085	0.4582	0.0668	4.951	5.220	4.0	7.0
	Accident risk level 3 soldier	26	5.038	0.8237	0.1615	4.706	5.371	3.0	7.0
	Subtotal	251	5.052	0.5876	0.0371	4.979	5.125	3.0	7.0

**Table 2 brainsci-13-01157-t002:** Test for homogeneity of variance.

	Levene Statistics	df1	df2	CTT Significance
Concentration Level	6.734	3	247	0.000
Brain activity	0.756	3	247	0.520
Psychological stress	1.577	3	247	0.195
Imbalance of left and right brain activity	0.453	3	247	0.715

**Table 3 brainsci-13-01157-t003:** Analysis of variance (ANOVA) results for EEG.

		The Sum of Squares	The Degree of Freedom	Mean Square	F	Significance Probability
Concentration Level	Between groups	62.947	3	20.982	6.093	0.001
	Within the group	850.628	247	3.444		
	The entire	913.574	250			
Brain activity	Between groups	956.733	3	318.911	5.495	0.001
	Within the group	14,335.156	247	58.037		
	The entire	15,291.888	250			
Psychological stress	Between groups	7.434	3	2.478	1.698	0.168
	Within the group	360.410	247	1.459		
	The entire	367.844	250			
Imbalance of left and right brain activity	Between groups	0.880	3	0.293	0.848	0.469
	Within the group	85.447	247	0.346		
	The entire	86.327	250			

**Table 4 brainsci-13-01157-t004:** EEG Multicomparison Results.

	Dependent Variable	Group (I)	Group (J)	Average Difference (I–J)	Standardization Error	Probability of Significance	95% Confidence Interval for the Average	
							Lower Limit	Upper Limit
Dunnett T3	Concentration level	General soldiers	Accident risk level 2 soldier	0.9062 *	0.2898	0.014	0.129	1.683
			Accident risk level 3 soldier	1.3859 *	0.2573	0.000	0.691	2.081
		Accident risk level 1 soldier	Accident risk level 3 soldier	1.3474 *	0.3659	0.003	0.355	2.340
Scheffe	Brain activity	General soldiers General soldiers	Accident risk level 1 soldier	−0.566	1.345	0.981	−4.35	3.22
			Accident risk level 2 soldier	3.753 *	1.289	0.039	0.12	7.38
			Accident risk level 3 soldier	4.742 *	1.631	0.040	0.15	9.33

* Mean differences are significant at the 0.05 level.

**Table 5 brainsci-13-01157-t005:** Technical Statistics.

		N	Average	Standard Deviation	Standard Error	95% for the Mean Confidence Interval		Minimum	Maximum
						Lower Limit	Upper Limit		
Heart health	General soldiers	136	13.7765	3.57091	0.30620	13.1709	14.3820	2.00	24.50
	Accident risk level 1 soldier	42	13.7271	3.73957	0.57703	12.5618	14.8925	6.30	23.50
	Accident risk level 2 soldier	47	12.3670	3.25648	0.47501	11.4109	13.3232	6.68	25.00
	Accident risk level 3 soldier	26	13.3885	4.11029	0.80609	11.7283	15.0486	5.85	24.00
	Subtotal	251	13.4641	3.62098	0.22855	13.0140	13.9142	2.00	25.00
Body stress	General soldiers	136	43.684	4.9166	0.4216	42.850	44.518	30.0	62.0
	Accident risk level 1 soldier	42	44.214	5.6935	0.8785	42.440	45.989	33.0	66.0
	Accident risk level 2 soldier	47	46.362	4.9142	0.7168	44.919	47.805	34.0	58.0
	Accident risk level 3 soldier	26	46.885	6.1535	1.2068	44.399	49.370	35.0	69.0
	Subtotal	251	44.606	5.3111	0.3352	43.945	45.266	30.0	69.0
Cumulative fatigue	General soldiers	136	4.809	0.4475	0.0384	4.733	4.885	2.0	5.0
	Accident risk level 1 soldier	42	4.857	0.3542	0.0546	4.747	4.968	4.0	5.0
	Accident risk level 2 soldier	47	4.681	0.4712	0.0687	4.543	4.819	4.0	5.0
	Accident risk level 3 soldier	26	4.731	0.5335	0.1046	4.515	4.946	3.0	5.0
	Subtotal	251	4.785	0.4489	0.0283	4.729	4.841	2.0	5.0
Physical Vitality	General soldiers	136	4.934	0.3031	0.0260	4.882	4.985	3.0	5.0
	Accident risk level 1 soldier	42	4.833	0.5372	0.0829	4.666	5.001	3.0	5.0
	Accident risk level 2 soldier	47	4.851	0.4159	0.0607	4.729	4.973	3.0	5.0
	Accident risk level 3 soldier	26	4.654	0.7452	0.1462	4.353	4.955	3.0	5.0
	Subtotal	251	4.873	0.4378	0.0276	4.818	4.927	3.0	5.0
Autonomic nervous system health	General soldiers	136	7.6746	0.75925	0.06510	7.5458	7.8033	5.68	9.17
	Accident risk level 1 soldier	42	7.6667	0.81587	0.12589	7.4124	7.9209	5.46	9.78
	Accident risk level 2 soldier	47	7.3209	0.83791	0.12222	7.0748	7.5669	5.63	8.79
	Accident risk level 3 soldier	26	7.5112	0.76082	0.14921	7.2039	7.8185	5.54	8.73
	Subtotal	251	7.5901	0.79162	0.04997	7.4917	7.6885	5.46	9.78

**Table 6 brainsci-13-01157-t006:** Test for homogeneity of variance.

	Levene Statistics	df1	df2	CTT Significance
Heart health	0.924	3	247	0.430
Body Stress	0.100	3	247	0.960
Cumulative fatigue	4.186	3	247	0.006
Physical Vitality	13.008	3	247	0.000
Autonomic nervous system health	0.205	3	247	0.893

**Table 7 brainsci-13-01157-t007:** Analysis of variance (ANOVA) results for Pulse Wave.

		The Sum of Squares	The Degree of Freedom	Mean Square	F	Significance Probability
heart health	Between groups	72.893	3	24.298	1.873	0.135
	Within the group	3204.976	247	12.976		
	The entire	3277.869	250			
Body Stress	Between groups	401.971	3	133.990	4.977	0.002
	Within the group	6649.981	247	26.923		
	The entire	7051.952	250			
Cumulative fatigue	Between groups	0.882	3	0.294	1.467	0.224
	Within the group	49.500	247	0.200		
	The entire	50.382	250			
Physical vitality	Between groups	1.841	3	0.614	3.289	0.021
	Within the group	46.080	247	0.187		
	The entire	47.921	250			
Autonomic nervous system health	Between groups	4.786	3	1.595	2.594	0.053
	Within the group	151.880	247	0.615		
	The entire	156.666	250			

**Table 8 brainsci-13-01157-t008:** Pulse Wave Multicomparison Results.

	Dependent Variable	Group (I)	Group (J)	Average Difference (I–J)	Standardization Error	Probability of Significance	95% Confidence Interval for the Average	
							Lower Limit	Upper Limit
Scheffe	Body stress	General soldiers	Accident risk level 1 soldier	−0.5305	0.9160	0.953	−3.109	2.048
			Accident risk level 2 soldier	−2.6779 *	0.8779	0.027	−5.149	−0.207
			Accident risk level 3 soldier	−3.2008 *	1.1106	0.042	−6.327	−0.075
Dunnett t	Physical vitality	General soldiers	Accident risk level 3 soldier	0.2008 *	0.0952	0.007	−0.066	0.494
		Accident risk level 1 soldier	Accident risk level 3 soldier	0.1795	0.1078	0.200	−0.070	0.429
		Accident risk level 2 soldier	Accident risk level 3 soldier	0.1970 *	0.1056	0.135	0.047	0.441

* Mean differences are significant at the 0.05 level.

**Table 9 brainsci-13-01157-t009:** EEG, pulse wave, maladjustment, and suicidal ideation index averages.

	Average	Standard Deviation	N
Concentration Level	6.235	1.9116	251
Brain activity	76.33	7.821	251
Psychological stress	5.946	1.2130	251
Imbalance of left and right brain activity	5.052	0.5876	251
Body Stress	44.606	5.3111	251
Cumulative fatigue	4.785	0.4489	251
heart health	13.4641	3.62098	251
Physical vitality	4.873	0.4378	251
Autonomic nervous system health	7.5901	0.79162	251
Maladjustment	2.2083	1.11388	240
Suicidal concerns	1.9269	1.13635	240

**Table 10 brainsci-13-01157-t010:** Correlation between EEG and pulse wave indices.

		Body Stress	Cumulated Fatigue	Heart Health	Physical Vitality	Autonomic Neuronal System Activity
Concentration Level	Pearson correlation	−0.102	0.051	0.062	0.066	0.134 *
	Probability of significance (both sides)	0.109	0.418	0.327	0.300	0.034
	N	251	251	251	251	251
Brain activity	Pearson correlation	−0.036	−0.039	0.038	−0.095	0.034
	Probability of significance (both sides)	0.569	0.542	0.545	0.134	0.588
	N	251	251	251	251	251
Psychological stress	Pearson correlation	−0.077	0.062	0.004	0.109	0.050
	Probability of significance (both sides)	0.221	0.331	0.951	0.085	0.432
	N	251	251	251	251	251
Imbalance of left and right brain activity	Pearson correlation	−0.042	0.073	−0.048	−0.021	0.034
	Probability of significance (both sides)	0.506	0.251	0.449	0.742	0.596
	N	251	251	251	251	251

* Correlation is significant at 0.05 level (both sides).

**Table 11 brainsci-13-01157-t011:** Correlating brain wave indices with survey results.

		Concentration Level	Brain Activity	Psychological Stress	Imbalance of Left and Right Brain Activity
Maladjustment	Pearson correlation	−0.203 **	−0.198 **	−0.101	−0.004
	Probability of significance (both sides)	0.002	0.002	0.117	0.951
	N	240	240	240	240
Suicidal concerns	Pearson correlation	−0.216 **	−0.229 **	−0.073	−0.004
	Probability of significance (both sides)	0.001	0.000	0.259	0.946
	N	240	240	240	240

** Correlation is significant at 0.01 level (both sides).

**Table 12 brainsci-13-01157-t012:** Correlate pulse wave index with survey results.

		Body Stress	Cumulative Fatigue	Heart Health	Physical Vitality	Autonomic Nervous System Health
Maladjustment	Pearson correlation	0.201 **	−0.088	−0.068	−0.182 **	−0.114
	Probability of significance (both sides)	0.002	0.172	0.296	0.005	0.078
	N	240	240	240	240	240
Suicidal concerns	Pearson correlation	0.231**	−0.108	−0.086	−0.131 *	−0.123
	Probability of significance (both sides)	0.000	0.095	0.184	0.043	0.057
	N	240	240	240	240	240

* Correlation is significant at 0.05 level (both sides). ** Correlation is significant at 0.01 level (both sides).

## Data Availability

Not applicable.

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
