# Peer review of "A Study of Brain Function Characteristics of Service Members at High Risk for Accidents in the Military"

_brainsci, 2023, doi:10.3390/brainsci13081157_

Round 1
Reviewer 1 Report (New Reviewer)
Although this is an interesting topic, a major revision is needed. The abstract does not follow the guidelines of the journal and does not provide enough details. The pictures need editing. In addition to that, the literature review is very poor. For example, relevant references should be added, regarding the fact that it is not clear if the included sample had prior history of Traumatic Brain Injuries (TBI) or sleep disorders or both which can act as moderators/mediators for the examined variables (see relevant research in athletes: Syrmos, N. C., Mylonas, A., Gavridakis, G., & Giannouli, V. (2017). Sleep disorders after mild traumatic brain injuries in elements with professional athletic activity. British Journal of Sports Medicine, 51(11), A36-A36.). The hypotheses should be presented as questions and supported by relevant references. The methodology is not clear to the reader regarding the procedures and the sampling. Which version of SPSS was used? Why are these statistical tests used instead of more complex analyses? The discussion should be more extended and needs support from relevant references.
Language editing is required.
Author Response
"첨부 파일을 참조하십시오."

Reviewer 2 Report (New Reviewer)
The present research article by Choi and colleagues, entitled ‘A Study on the Relationship between the Military Life Adaptation Degree of Soldiers, Brain Function and Autonomous Nervous System’ focuses on the investigation of brain waves and pulse waves in soldiers to analyze variations in brain function and the autonomic nervous system. The study aims to identify potential differences between general soldiers well-adjusted to military life and soldiers at risk of accidents. The research background highlights the negative consequences of military accidents on morale and public trust, despite existing prevention efforts. The urgent need for effective accident prevention measures is emphasized due to the increasing number of annual casualties. The manuscript discusses the characteristics of the military organization, stress, depression, and their impacts on accidents. Internal conflicts within the military organization, caused by cultural differences and undemocratic practices, are identified as factors that weaken combat power and contribute to accidents. The analysis of brain waves reveals notable variations among soldier groups. The concentration index and brain activity index show significant differences, while the brain stress index and bilateral brain balance index do not. The post-test results further confirm significant differences between specific groups, indicating distinct brain function patterns. The manuscript concludes with the significance of the findings and their implications for developing effective accident prevention measures. The study employed EEG and pulse wave measurement equipment to collect data from the study participants. The demographic characteristics of the subjects are presented, emphasizing the age and academic backgrounds of the soldiers. The manuscript highlights the general characteristics of the research subjects, providing insights into the sample population.
In general, I think the idea of this article is really interesting and the authors’ fascinating observations on this timely topic may be of interest to the readers of Brain Sciences. However, some comments, as well as some crucial evidence that should be included to support the author’s argumentation, needed to be addressed to improve the quality of the manuscript, its adequacy, and its readability prior to the publication in the present form, in particular reshaping parts of the Introduction and Methods sections by adding more evidence and theoretical constructs.
Please consider the following comments:
• I suggest changing the title. In my opinion, it should be more concise and straightforward, ensuring clarity for readers. Consider rephrasing it to convey the main idea more succinctly.
• Abstract: According to the Journal’s guidelines, the abstract should be up to a total of about 200 words maximum. In my opinion, Authors should consider rephrasing this section. According to the Journal’s guidelines, the Abstract should contain most of the following kinds of information in brief form. Please, consider giving a more synthetic overview of the paper's key points: I would suggest rephrasing the results and conclusion to make them clear for readers to understand.
• A graphical abstract that will visually summarize the main findings of the manuscript is highly recommended.
• In general, I recommend authors to use more references to back their claims, especially in the Introduction of this article, which I believe is lacking. Thus, I recommend the authors to attempt to expand the topic of their article, as the bibliography is too concise. Nevertheless, I believe that less than 60/70 articles are too low for a research article. Therefore, I suggest the authors to focus their efforts on researching relevant literature: in my opinion, adding more citations will help to provide better and more accurate background to this study.
• Introduction: In this section, the authors have presented a comprehensive overview of previous research on stress, depression, and their impact on decision-making and accident rates. However, to further enrich the discussion and enhance the readers' understanding, it would be beneficial to include information on the neural substrates underlying these phenomena. Given the growing body of literature on the neural correlates of stress and depression, incorporating details about the specific brain regions and pathways involved in regulating emotions and cognitive processes could provide a more nuanced perspective. For example, studies have implicated the prefrontal cortex, amygdala, and hippocampus in stress-related responses and decision-making, while highlighting the role of neurotransmitters such as serotonin and dopamine in modulating mood and cognitive functions (https://doi.org/10.3390/ijms24065926). Therefore, I would suggest to include information on neural substrates to strengthen the theoretical framework of the study but also to help bridge the gap between brain function and behavioral outcomes (DOI: 10.3390/ijms24044114; https://doi.org/10.1016/j.neubiorev.2023.105163). By elucidating the neurobiological mechanisms underlying stress-related accidents, the findings could have broader implications for intervention strategies and the development of targeted prevention programs.
• Methods: This section lacks detailed information about the sample selection process, inclusion/exclusion criteria, and potential biases. Additionally, while the Authors mention selecting soldiers from various military departments and locations, they do not provide a clear rationale for this selection process. It would be helpful to explain why these specific units and locations were chosen and how they represent the broader population of soldiers.
• Results: I suggest rewriting this section more accurately. To properly present experimental findings, I think that authors should provide full statistical details (like degree of freedom or post-hoc utilized), to ensure in-depth understanding and replicability of the findings. Also, here the Authors focus on identifying significant differences between groups but do not address the magnitude of these differences. Including effect sizes (e.g., Cohen's d) would provide a more comprehensive understanding of the practical significance of the observed group differences. Finally, the EEG index analysis section could benefit from a clearer presentation of the results, including tables or figures to illustrate the findings. Additionally, the post-test analysis should be explained in more detail.
• In my opinion, the ‘Conclusions’ paragraph would benefit from some thoughtful as well as in-depth considerations by the authors, because as it stands, it lists down all the main findings of the research, without really stressing the theoretical significance of the study. Authors should make an effort, trying to explain the theoretical implication as well as the translational application of their research.
• In according to the previous comment, I would ask the authors to include a proper and defined ‘Limitations and future directions’ section before the end of the manuscript, in which authors can describe in detail and report all the technical issues brought to the surface.
• Tables and Figures: According to the Journal’s guidelines, please provide a short explanatory caption for the table within the text. Also, I suggest to modify all figures for clarity because, as it stands, the readers may have difficulty comprehending it and to change the scale of the vertical axis and use the same minimum/maximum scale value in all the graphs. Also, please provide an explanatory caption for each table/figure within the text.
• Finally, the manuscript does not clearly highlight the novelty or significance of the study. I would suggest the authors to explicitly state the contribution of their research to the existing literature and explain how their findings advance the field.
I hope that, after these careful revisions, this paper can meet the Journal’s high standards for publication.
I am available for a new round of revision of this paper. I declare no conflict of interest regarding this manuscript.
Best regards,
Reviewer
Minor editing of English language required.
Author Response
"Please see the attachment."

Round 2
Reviewer 2 Report (New Reviewer)
Dear Authors,
I would like to express my appreciation for your insightful study on stress-related accidents. This study shed light on the complex interplay between stress, accidents, and behavioral outcomes. However, I believe that further enriching your work by providing more information on the neural substrates underlying these phenomena could greatly enhance the theoretical framework and overall impact of the study.
Given the growing body of literature on the neural correlates of stress and depression, incorporating details about the specific brain regions and pathways involved in regulating emotions and cognitive processes would provide a more nuanced perspective. For instance, I still recommend discussing the role of prefrontal cortex, amygdala, and hippocampus in stress-related responses and decision-making (DOI: 10.3390/ijms24044114; https://doi.org/10.1016/j.neubiorev.2023.105163).
By including information on neural substrates, your study would not only strengthen its theoretical foundation but also bridge the crucial gap between brain function and behavioral outcomes. Elucidating the neurobiological mechanisms underlying stress-related accidents has the potential to contribute to the development of targeted intervention strategies and prevention programs. Moreover, this deeper understanding of the neural underpinnings could have broader implications for improving safety measures and mitigating the impact of stress-related incidents.
Overall, this is a timely and needed work. It is well researched and nicely written, with a good balance between descriptive and narrative text. I just need to do one more comment, to provide a clearer and more understandable neural background to the readers, before continuing with the publication.
Best regards,
Reviewer
Author Response
"첨부 파일을 참조하십시오."

This manuscript is a resubmission of an earlier submission. The following is a list of the peer review reports and author responses from that submission.
Round 1
Reviewer 1 Report
The manuscript has to be edited and modified in order to make clear both novelty and motivation as well as the methodological concept. So, my suggestions are follow:
1) ABSTRACT has to be enhanced. Several sentences are merged to each other. what is the main finding of the present study? Is there a brief review on past studies ? If so, what is the most important issue of the study?
2) INTRODUCTION has to be considerably extended in order to mention the close relations between psychological states of the individuals and their behavioral/body functions. They should mention the neural mechanism of the brain in accordance with cognitive and behavioral neuroscience papers such as doi:10.1016/j.bspc.2022.103740 . Then, the motivation of the study should be clarified.
3) The sub-sections of INTRODUCTION should be moved into subsections of another main section so called 'METHODS AND MATERIALS'
4) The authors should follow international terminology in understanding brain functions. Thus, EEG sub-bands should be mentioned as frequency components of EEG recordings. Neuroelectrical brain activities can be recorded from scalp surface in terms of surface EEG. The origin of them is generated by both neurotransmitter activities and post-synaptic potentials. Characteristic properties of brain waves, i.e. EEG series are effected by not only cognitive but also external stimuli/events/conditions. The authors should be careful in defining the sub-bands. These frequency components can not be referred by uncommon symbols used in the manuscript. Please use the international definitions of them in form: delta, theta, alpha, beta, gamma
5) what does mean ''Degree of brain activity'' ? Please explain its meaning clearly. The authors should use common terminology such as 'degree of EEG complexity', 'relative power ratio', 'power spectral density estimations', 'functional state (sleep, awake, emotional, auditory, visual, cognitive, resting-state)', ...etc
6) what does mean 'Left-right brain balance' ? Do they mean 'inter-hemispheric asymmetry' or 'cross-correlations between right and left hemispheres' or 'EEG synchronization' ? Please clarify this concern.
7) Re-name Section 4.1. as 'Group Design and Participants'
8) Re-name Section 4.2. as a main section '5. Data Collection and Analysis Approach' and then write the concept of corresponding sub-sections 4.3. into this main section.
9) Re-name the section 5. as '6. Results'. Then, the corresponding subsections should be merged in Section6.
10) The whole manuscript has to be edited in accordance with both international terminology and English grammar in use.